# Ecological risk assessment and source identification of heavy metal pollution in vegetable bases of Urumqi, China, using the positive matrix factorization (PMF) method

Mireadili Kuerban[1,2], Balati Maihemuti[1,3]*, Yizaitiguli Waili[1], Tuerxun Tuerhong[4]

**1** College of Resources and Environmental Science, Xinjiang University, Urumqi, China, **2** College of Resources and Environmental Sciences, China Agricultural University, Beijing, China, **3** Key Laboratory of Xinjiang General Institutions of Higher Learning for Smart City and Environment Modeling, Xinjiang University, Urumqi, China, **4** College of Grassland and Environmental Science, Xinjiang Agricultural University, Urumqi, China

☯ These authors contributed equally to this work.

* bmaihemuti@xju.edu.cn

**Data Availability Statement:** All relevant data are within the paper and its Supporting Information files.

## Abstract

Heavy metal pollution is a widespread problem and strongly affects human health through the food chain. In this study, the overall pollution situation and source apportionment of heavy metals in soil (Hg, Cd, As, Pb, Ni, Zn, Cu and Cr) were evaluated using various methods including geo-accumulation index ($I_{geo}$), potential ecological risk index (RI) and positive matrix factorization combined with Geographical Information System (GIS) to quantify and identify the possible sources to these heavy metals in soils. The results of $I_{geo}$ showed that this farmland top soil moderate contaminated by Hg, other selected elements with noncontamination level. And the average RI in the top soil was 259.89, indicating a moderate ecological risk, of which Hg and Cd attributed 88.87% of the RI. The results of the PMF model showed that the relative contributions of heavy metals due to atmospheric depositions (18.70%), sewage irrigations (21.17%), soil parent materials (19.11%), industrial and residential coal combustions (17.43%) and agricultural and lithogenic sources (23.59%), respectively. Of these elements, Pb and Cd were came from atmospheric deposition. Cr was attributed to sewage irrigations. As was mainly derived from the soil parent materials. Hg originated from industrial and residential coal combustions, and most of the Cu, Zn and Ni, except for Pb, were predominantly derived from agricultural and lithogenic sources. These results are important in considering management plans to control the aggravation of heavy metal pollution and ultimately to protect soil resources in this region. In addition, this study enhances the understanding of heavy metal contamination occurrence in agroecosystem that helps predicting and limiting the potential of heavy metal exposure to people and ecosystem.

**Funding:** This study was supported by the Natural Science Foundation of Xinjiang Uygur Autonomous Region of China and supported by the National Natural Science Foundation of China (Grant No. 41762019).

**Competing interests:** The authors declare that there is no conflict of interests regarding the publication of this paper.

## 1. Introduction

The accumulation of heavy metals in soils not only leads to a decline in the production and quality of agricultural yield but also poses a serious threat to human health through the food chain, as their detrimental impact appears after several years of exposure [1]. Thus, heavy metal pollution directly influences the quality and safety of agriculture products by affecting the soil environmental quality and safety [2–3], which is not only key to the sustainable development of farmland resources and land conservation but also the basis of national food security. Soils are vulnerable and recover with much difficulty from environmental contamination because, although slow auto-remediation processes are implemented, the fast dispersal and dilution mechanisms meet functional limitations in soils [4]. Heavy metal (Hg, Cd, As, Pb, Ni, Zn, Cu and Cr) accumulations in farmland soils are caused by the contamination of agricultural lands and deterioration of the environment, possibly due to the long-term toxicity, strong latency, and low migration rate [5]. However, certain heavy metals (Cu and Zn) are critical for plants and living organisms up to a certain content. They might become harmful when their concentration exceeds the primary value, and toxic effects are likely to occur and to pose a threat when heavy metals enter the human body via the food chain [6–9]. The concentration of heavy metal elements in the soil is an important indicator of the soil environmental quality in vegetable bases [10]. Furthermore, the high level of accumulation of heavy metals in vegetable fields not only directly changes the physical and chemical properties of the soil but also leads to the decline in the vegetable quality and variety [11]. Such effects are likely to bring about potential risks to both human health through the food chain and environmental quality and safety through secondary pollution [12]. Therefore, heavy metals are persistent and accumulative, which can pose potential risks to ecosystem and human health [13–14]. Ecological risk assessment is an effective tool to evaluate the impact of chemical contaminants on ecosystems [15]. In this regard, the objective of our study is to present and discussed properly for the first time the ecological risk that could be associated to heavy metals in surface soils of this vegetable bases. Then we using the positive matrix factorization (PMF) method to fully identify the possible different pollution sources and relative contributions of the eight heavy metals.

To control and prevent heavy metal pollution, the source identification and apportionment are very important, and the selection of a proper and effective model is essential for accurate results [16–17]. Several receptor models have been used to identify heavy metal sources. The models are principal component analysis (PCA), unmix models (UNMIX), chemical mass balance (CMB)and positive matrix factorization (PMF) model. All models have their upsides and downsides, as demonstrated in previous comparison studies [18–19]. PMF model is a well-known receptor model that along with the combination of multivariate statistics, has been widely used for apportioning the source of heavy metals. Compared with the other three models, the factors obtained from the PMF analysis represent the main sources that were used to yield the simulated data most closely. The short non-negative constraint is another remarkable downside of the PCA, APCS and CMB methods [20]. Many studies were carried out using the PMF model and valuable results were obtained [1], [21–24].

The aims of this study to determine the present state of heavy metal pollution and the lateral ecological risk of heavy metals as well as to determine the possible contamination sources in the suburban vegetable bases in Urumqi, China, to provide a scientific basis for the prevention and control of pollution, promote the production of green vegetables and ensure the quality and safety of the vegetables for protecting local human health. Urumqi is an economically quickly developing inland city with a permanent population of 3.5 million inhabitants in north western arid China [25]. However, in 1998, an evaluation by the World Health Organization (WHO) indicated that Urumqi is one of the top 10 (ranked as the fourth) most heavily polluted

cities in the world [26–27]. Thus, it is essential to conduct research on the suburban vegetable planting area. A previous study [28] focused on analysing the health risk assessment and pollution characteristics of six heavy metals on this vegetable basis but did not specifically perform heavy metal source apportionment. In this paper, we use the geo-accumulation index ($I_{geo}$), potential ecological risk index (RI) to evaluate the present pollution states and potential risks and use the positive matrix factorization (PMF) method to fully identify the possible different pollution sources and relative contributions of the eight heavy metals. The results obtained from this study provide both scientific insights for the further control and prevention of heavy metal contamination in suburban agriculture areas and an objective basis for safe consumption.

## 2. Materials and methods

### 2.1 Study area

Urumqi, located in the Xinjiang oasis, is the capital of the Xinjiang Uygur Autonomous Region and a typical inland metropolitan city in the northwest region of China. Urumqi (Fig 1) (approximately located between 86° 37′ 33″ and 88° 58′ 24″ E, and between 42° 45′32″ and 44° 08′ 00″ N) is surrounded by the northern foot of the Tianshan Mountains and the Junger Basin to the north, with a temperate continental climate.

The cultivation area of vegetables in Urumqi includes the northern vegetable bases (Anningqu Town) and the southern bases. The Anningqu Town, situated in the northern suburbs of Urumqi, is a triangular area, where 312 National Highway, 216 National Highway, TuWuDa Highway and Wukui Highway intersect, with an area of approximately 120 km$^2$ [28]. Additionally, the main products of the area are tomatoes, beans, wheat, radish, bitter gourd, and cabbage; finally, groundwater or drainage water is used for irrigation in this area.

### 2.2 Sampling

There were 146 soil samples in total collected at a depth of approximately 0–20 cm from the Anningqu Town of Urumqi during July 2017 for the study (Fig 1). Soils in the vegetable

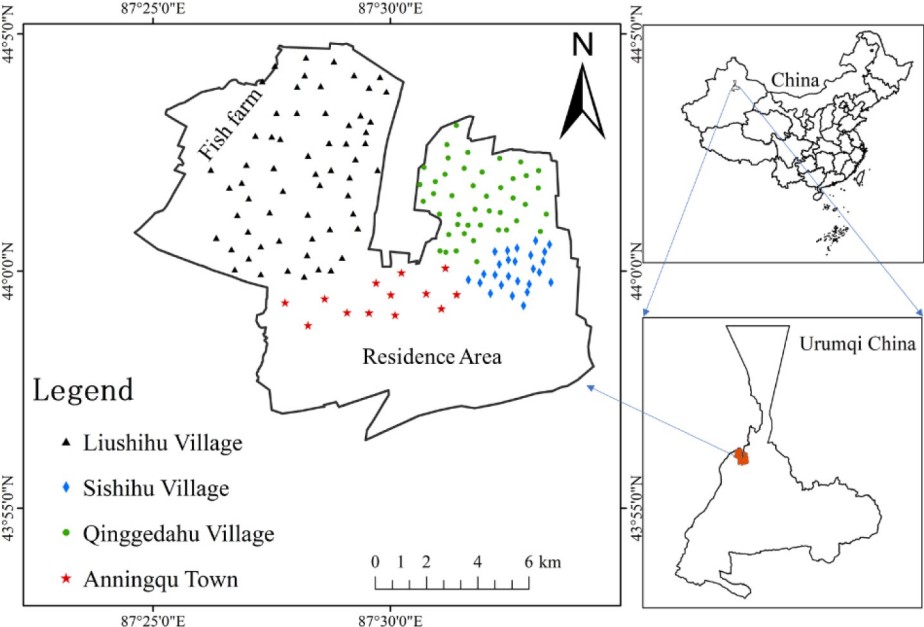

**Fig 1. Location of study area and sampling sites.**

farmland is light loam and light sandy with pH ranging from 7.9 to 8.0. The majority of the selected sample sites (56 out of 146; 38.36%) was located in Liushihu village, while the rest of the sites were located in Qinggedahu village (42 out of 146; 28.77%), Sishihu village (26 out of 146; 17.81%) and Anningqu town (22 out of 146; 15.07%). To ensure the soil sample quality control, the fieldwork was performed based on a standard operation procedure (SOP). Considering the characteristics of the topography and planting area, the soil sampling was conducted via the grid method with a 0.7 km × 0.7 km grid, while the sampling point locations were recorded using the global positioning system (GPS). Additionally, approximately 3 to 5 subsamples were taken at each grid point, randomly mixed and the quartile method was used to obtain a bulk sample of approximately 1.0kg. Finally, the bulk samples were stored in polyethylene bags, which were transported to the Xinjiang University laboratory.

## 2.3 Sample processing

The soil samples were air-dried in the laboratory with the methods of the Environmental Protection Standards of the People's Republic of China (HJ 803–2016) issued by the Ministry of Environmental Protection, and then, the samples, which had been dried, were sieved with a sieve that had a sieve size of ≤0.149 mm. Soil pH was determined in soil and water of 1:2.5 (w/v), using a pHS-3C digital pH meter (Shanghai REX Sensor Technology Co., Ltd., China) in accordance with the agricultural sector standard of People's Republic of China (NY/T1377-2007). Soil texture was determined by a laser particle size analyzer. The concentration of soil organic matter (SOM) in farmland were tested in the Xinjiang University laboratory used SOM fractionation method. Thereafter, 0.25g of the soil samples was placed in a 50ml Teflon Crucible and digested using the $HN0_3$-$HClO_4$-HF-HCl digestion method on a hot plate. Finally, the total As and Hg concentrations were measured by a Beijing General Analytical Instrument Co. PF6-2 dual channel automatic atomic fluorescence spectrometer, and the detection limits for Hg and As were 0.005 and 0.01 mg/kg, respectively. The total Zn, Cu, Cr, Pb, Pb, Cd and Ni concentrations were determined using an atomic absorption spectrophotometer. The detection limits for the heavy metals Zn, Cu, Cr, Pb, Cd, and Ni were 0.5, 1.0, 2.5, 0.06, 0.05 and 2.5 mg/kg, respectively. To ensure the accuracy of the analysis, the GSS-12 method (with geochemical soil standard references samples) was adopted for the purpose of quality control, and each sample was subjected to three replicates of parallel experiment treatment, and the mean value was used for analysis.

## 2.4 Pollution assessment methods

**2.4.1 Geo-accumulation index.** To evaluate the heavy metal contamination level, the geo-accumulation index proposed by Müller [29] was used in this experiment.

$$I_{geo} = log_2\left(\frac{C_i}{1.5 \times B_i}\right) \qquad (1)$$

where $I_{geo}$ is the geo-accumulation index of a sample site; $C_i$ is the measured concentration of heavy metal $i$ in the soil, mg/kg; $B_i$ is the background value of heavy metal $i$, mg/kg; and 1.5 is the background matrix correction factor due to lithospheric effects. In this study, the soil background values of Xinxiang were used as references to assess the present pollution state and potential for ecological risks, and the background values for Hg, Cd, As, Pb, Ni, Zn, Cu and Cr were 0.017, 0.12, 11.2, 19.4, 25.20, 68.8, 26.70 and 49.3 mg/kg, respectively (CSEPA, 1990). The classifications of $I_{geo}$ are: $I_{geo} \leq 0$ is no contamination (I), $0 < I_{geo} \leq 1$ is light to moderate (II), $1 < I_{geo} \leq 2$ is moderate (III), $2 < I_{geo} \leq 3$ is moderate to heavy (IV), $3 < I_{geo} \leq 4$ is heavy (V), $4 < I_{geo} \leq 5$ is heavy to extremely serious (VI) and $I_{geo} \geq 5$ is extremely serious (VII), respectively.

**2.4.2 Potential ecological risk assessment.** To assess the level of ecological risks, potential ecological risk index (RI) methods were used, which were proposed by Hankinson [30], according to the characteristics of the heavy metals and their environmental behaviour. The RI is highly associated with three coefficients, namely, the individual pollution coefficient, the response coefficient of heavy metal toxicity and the potential ecological risk individual coefficient, and can be expressed as follows [31]:

$$\text{RI} = \sum_{i=1}^{n} E_j^i = \sum_{i=1}^{n} (T^i \times C_j^i) = \sum_{i=1}^{n} (T^i \times \frac{C_i}{B_i}) \tag{2}$$

where RI is the potential ecological risk index, $E_j^i$ is the potential ecological risk individual coefficient of heavy metal $i$ at sample site $j$, and $T^i$ is the toxicity response coefficient of heavy metal $i$. In this study, we adopted reference toxicity values for each heavy metal in the order of $T^{Zn} = 1$, $T^{Cr} = 2$, $T^{Cu} = T^{Ni} = T^{Pb} = 5$, $T^{As} = 10$, $T^{Cd} = 30$ and $T^{Hg} = 40$. $C_i$, $B_i$ and n followed the same order as above. The classification conditions of potential ecological risks are shown in Table 1 [32].

**2.4.3 Positive matrix factorization (PMF) model.** The positive matrix factorization (PMF) model is a multivariate receptor model that uses pollution source identification because a PMF model requires no source profiles, uses uncertainty-weighted data and a non-negativity constraint never occurs with PMF modelling [33]. The identifying results from PMF modelling provide better explanations than the other methods, such as principal component analysis (PCA). Thus, in this study, we used the PMF model to identify the contamination source of the heavy metals.

The calculation process via a PMF model is to factorize the original matrix $E_{ik}$ into two factor matrices, $X_{ij}$ and $Y_{jk}$, as well as a residual matrix $Z_{ik}$, which is shown as follows:

$$E_{ik} = \sum_{j=1}^{p} X_{ij} \cdot Y_{jk} + Z_{ik} \quad (i = 1, 2, \ldots, n; \; k = 1, 2, \ldots, m)$$

Where $E_{ik}$ is the concentration of the kth heavy metals in the ith sample; $X_{ij}$ is the contribution of the jth heavy metal on the $i$th sample; and $Y_{jk}$ is the factorization of the jth heavy metal that is adjacent to heavy metal k. $X_{ij}$ (the factor contributions) and $Y_{jk}$ (the factor profiles) were derived from the PMF receptor model by minimizing the objective function Q, as shown below [34]:

$$Q = \sum_{i=1}^{n} \sum_{k=1}^{m} \left(\frac{Z_{ik}}{t_{ik}}\right)^2$$

where $t_{ik}$ is the uncertainty of the $k$ th heavy metal for the $i$th sample. If the heavy metal concentration is higher than the minimum detection limit (MDL), which is calculated using: Unc

**Table 1. Classification criteria of potential ecological risk index.**

| Grades | $I_{geo}$ | $E_j^i$ | RI | Class of ecological risk |
|--------|-----------|---------|-----|--------------------------|
| I | $I_{geo} \leq 0$ | $E_j^i < 40$ | RI < 110 | Low potential ecological risk |
| II | $0 < I_{geo} \leq 1$ | $40 \leq E_j^i < 80$ | $110 \leq RI < 220$ | Moderate potential risk |
| III | $1 < I_{geo} \leq 2$ | $80 \leq E_j^i < 160$ | $220 \leq RI < 440$ | Considerable potential ris |
| IV | $2 < I_{geo} \leq 3$ | $160 \leq E_j^i < 320$ | $440 \leq RI < 880$ | High potential risk |
| V | $3 < I_{geo} \leq 4$ | $E_j^i \geq 320$ | $800 \leq RI$ | Significantly very high |

= $[(\text{error fraction} \times \text{concentration})^2 + (MDL)^2]^{1/2}$; otherwise, is calculated using: Unc = 5/6×MDL, where Unc represents the uncertainty [1].

**2.4.4 Statistical analysis.** SPSS 19.0 and Microsoft Excel 2010 were used to perform the data analysis. ArcGIS 10.2.2 software (ESRI, US) was used to map the sampling sites. The heavy metal source analysis was conducted using a positive matrix factorization [34] analysis model. Origin (8.5) was used to map the index of geo-accumulation for the vegetable bases as well as the percentages of sites at different pollution levels among the total sample sites, potential ecological risk assessment results and ecological risk warning assessment results.

## 3. Results and discussions

### 3.1 Concentration of heavy metals

The soil types in the study area are black soil, sandy soil and clayey soil, and the soil texture is mainly silt loam but also contains sand and clay in small percentages with pH ranging from 4.79 to 7.25, average of the pH values is 6.56. The concentration of SOM in farmland is between 4.39 and 31.21g /kg, with an average of 10.89g/kg. Heavy metals in soil of the vegetable bases showed spatial and element-specific variety (Table 2). Mean concentrations of Cu, Zn, Ni, Pb, Cd, Hg, As and Cr were 34.88, 94.44, 33.68, 22.07, 0.17, 0.08, 6.89 and 61mg/kg, respectively. Overall, an average concentration of the heavy metal, except for As, were obviously greater than their background values in Xinjiang. The soil environmental quality standard is mainly used to guarantee and protect agricultural land and human health, thus heavy metal contents in soil exceeding the corresponding secondary criteria provide significant basis for determining the harm to human health. While the mean concentrations of every heavy metal in soil have corresponding secondary standards, the maximum contents of As and Cd were exceeded the secondary criteria, which indicated that As and Cd in the vegetable bases obviously accumulated. In fact, there is a strong focus on Cd in Chinese agricultural soils with intensive monitoring to prevent further accumulation. The CV values were calculated for the eight heavy metals because this value demonstrates the average variation degree for each sample. The CV values for Hg and As were 96.20 and 110.16%, respectively, which indicated a high variation. In contrast, the CV values of the other six heavy metals were below 40% (Cu at 23.16, Zn at 16.70, Ni at 19.79, Pb at 28.73, and Cd at 39.08%), indicating that those heavy metals had moderate to little variation (showed in Table 2).

**Table 2. Statistical summary of heavy metal concentrations in vegetable bases (mg/kg).**

| Heavy metals | Cu | Zn | Ni | Pb | Cd | Hg | As | Cr |
|---|---|---|---|---|---|---|---|---|
| Mean | 34.88 | 94.44 | 33.68 | 22.07 | 0.17 | 0.08 | 6.89 | 61.00 |
| SD [a] | 8.08 | 15.77 | 6.66 | 6.34 | 0.07 | 0.07 | 7.59 | 15.39 |
| Minimum | 18.94 | 63.75 | 15.58 | 5.50 | 0.06 | 0.01 | 0.01 | 20.84 |
| Maximum | 63.69 | 179.05 | 59.60 | 38.42 | 0.66 | 0.46 | 34.26 | 103.62 |
| CV [b] (%) | 23.16 | 16.70 | 19.79 | 28.73 | 39.08 | 96.02 | 110.16 | 25.24 |
| Skewness | 1.02 | 2.05 | 0.33 | 0.24 | 3.15 | 1.72 | 1.21 | 0.22 |
| Kurtosis | 1.60 | 8.33 | 1.52 | -0.17 | 18.79 | 4.08 | 1.62 | 0.03 |
| Background value [c] | 26.70 | 68.80 | 25.20 | 19.40 | 0.12 | 0.017 | 11.20 | 49.30 |
| Chinese soil criteria [d] | 100 | 300 | 60 | 350 | 0.6 | 1 | 25 | 250 |

[a] SD means standard deviation.

[b] CV means coefficient of variation.

[c] Soil heavy metal background value of Xinjiang.

[d] Soil environmental quality standard (GB15618-1995).

The results from several former researchers' studies were listed in the S1 Table. Obviously, concentration of Zn, Cu, Ni, Cd, Pb and Cr for 12 suburban agricultural land ranged from 21.22~190.4 mg/kg, 14.02~101.25 mg/kg, 20.70~93.70mg/kg, 0.1347~2.10mg/kg, 10.56~66.10mg/kg, 37.04~97.00mg/kg respectively; Concentration of Hg and As for 5 suburban farmland ranged from 0.079~0.13mg/kg and 6.16~11.20mg/kg respectively. Content of Zn, Cu and Cr in this study area ranked to fifth, which indicated that those heavy metals stayed up middle accumulation level. By comparison, Ni content in this study area was higher than inland suburban areas, but lower than European city (Lisbon) and Middle East city (Tabriz). The Pb content in this suburban area compared with 7 inland suburban farmland areas exception of one suburban farmland area (Taihang Piedmont Plain) and two foreigner suburban farmland area exception of one suburban farmland area (Tabriz city), however, were lower in some extent. While concentration of Cd in this pre-urban agricultural land ranked seventh, this value exceeded references value that used in this study. In comparison, Hg content in this study area was lower than other four inland suburban. The difference of Hg content was not obvious compare to Nanjing city and Taihang Piedmont Plain, while concentration of Hg in this study area was lowest. By contrast, concentration of As in this study area was lower than suburban farmland of Nanjing, Beijing and Xianyang city respectively, and slightly higher than suburban farmlands of Taihang Piedmont Plain [35–40].

## 3.2 Environmental quality evaluation of the vegetable base soils

The box chart of the geo-accumulation index ($I_{geo}$) for each metal in the soil are shown in Fig 2A. The mean index values of the heavy metals were ranked in the order of Hg>Cd>Zn>Ni>Cu>Cr>Pb>As; all of the mean index values of the heavy metals were lower than zero, apart from that of Hg (1.03), which showed that the soil of the vegetable basis was not polluted with Cd (-0.13), Zn (-0.15), Ni (-0.20), Cu (-0.24), Cr (-0.33), Pb (-0.44) nor As (-0.384). The mean $I_{geo}$ value of Hg was greater than one, which indicated that Hg was the main heavy metal that resulted in vegetable basis contamination with heavy metals, and its contamination degree was light to moderate contamination. The percentages of sites at different pollution levels among the total sample sites are shown in Fig 2B. While the mean $I_{geo}$ value of Zn was higher than the mean index value of Cu, the percentages of uncontaminated

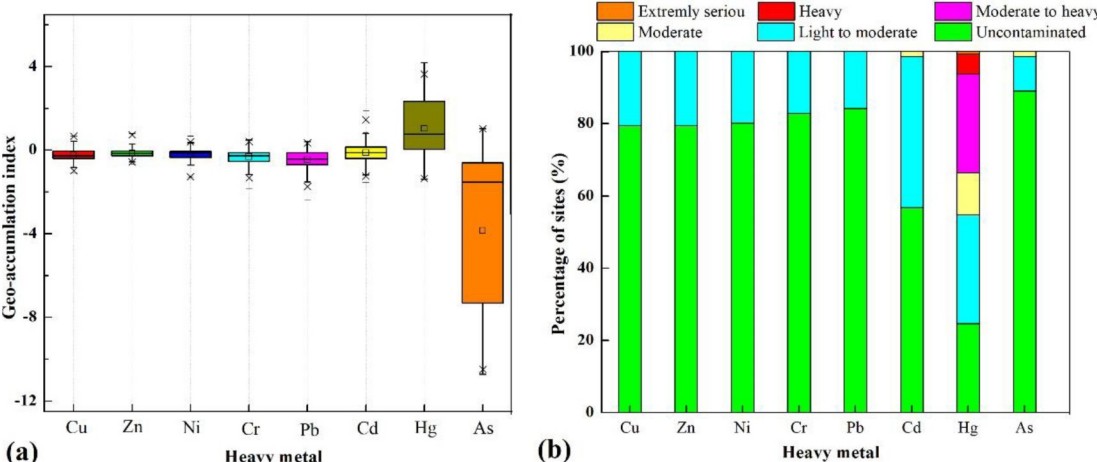

**Fig 2. Results of eight heavy metal geo-accumulation index.** (a. The black line and bar, lower and upper edges, bars and forks in or outside the boxes represent median and mean values, 25th and 75th, 5th and 95th percentiles of all data, respectively; b. Percentage of sites at different pollution level).

and lightly to moderately contaminated sample sites were the same for both Zn and Cu at 79.45%(uncontaminated) and 20.55% (lightly to moderately contaminated), respectively. The heavy metals (Cu, Zn, Ni, Cr and Pb) exhibited the same contamination degree, of which the percentage of sample sites that were not contaminated with Pb (84.25%) was the highest among the five heavy metals, and the percentage of Zn- and Cu-contaminated sample sites was larger compared to the other three heavy metals. The pollution degree and percentage of moderately contaminated sample sites were the same for both Cd and As, but the ranking of the $I_{geo}$ values of all the heavy metals was very different owing to the percentage of sites at different pollution levels among the total sample sites. The percentage of sites at different pollution levels of Hg differed from that of the other heavy metals, and the Hg contamination degree reached extremely serious levels, and those contamination levels decreased according to the following order: light to moderate (30.14%) > moderate to heavy (27.40%) > uncontaminated (24.66%) > moderate (11.64%) > heavy(5.48%) > extremely serious (0.68%).

## 3.3 Ecological risk assessment

The potential ecological risk assessment for the individual heavy metals was calculated, as described in Fig 3A and classification criteria of potential ecological risk index showed in Table 1. The mean values of the potential risk coefficient ($E_j^i$) of each heavy metal were ranked in the order of Hg>Cd>Ni>Cu>As>Pb>Cr>Zn, with values of 187.39, 43.6, 6.68, 6.53, 6.16, 5.46, 2.47 and 1.37, in respectively. The $E_j^i$ values of Ni, Cu, As, Pb, Cr and Zn were lower than 40, and all belonged to the low risk level (Table 1). The $E_j^i$ value for Cd was higher than 40 but lower than 80; thus, Cd exhibited a moderate risk for the vegetable basis. The potential ecological risk of Hg reached the heavy risk level, which indicated that Hg had a high potential ecological risk for the vegetable bases and was the main element that caused potential ecological risk. The differences between the potential ecological risk index and the geo-accumulation index are reflected in the results of Cu, Zn, Ni, Cr, Pb, Cd and As being categorized as light pollution. While the mean $I_{geo}$ values of those metals indicated no contamination, the potential risk index showed that there was a low contamination risk for Cu, Zn, Ni, Cr, Pb and As, as well as a moderate contamination risk for Cd. The reason is mainly that the geo-accumulation index method emphasized the comparative evaluation of the heavy metal concentrations in the

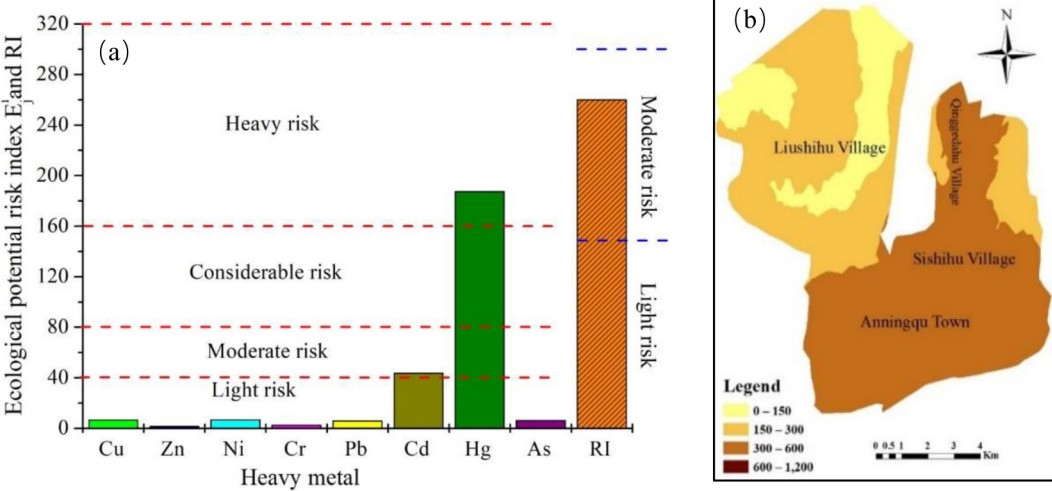

**Fig 3. Environmental quality evaluation of vegetable bases soils.** a. Results of potential ecological risk assessment; b. Spatial distribution of the RI in the vegetable bases.

soil, but the potential ecological risk index focused on the toxic differences of the heavy metal elements. In the entire study area, the mean value of RI in the soil was 259.89, which indicates that all sample sites on a vegetable basis belonged to a moderate potential ecological risk level.

The spatial variation of the potential ecological risk assessment (RI) of heavy metals in the vegetable bases is presented in Fig 3B. The RI concentration results showed that the spatial distribution of RI had a notable zonal distribution pattern. The largest value of RI (approximately 220–440) was distributed mainly in Anningqu town and Sishihu village as well as in half of the area of Qinggedahu village. Anningqu town and Sishihu village were not only closely located to urban areas but also had the highest density of human activities, specifically transportation and industrial activities (steel, cement, metallurgy, processing, etc.). Qinggedahu village was closely located to the Midong industrial area and used groundwater and the effluent from a sewage treatment plant for irrigation. The lowest and moderate values of RI (Table 1) were mainly observed in Liushihu village (Fig 3B). Due to it's small population size there is less human impacts on environmental pollution, furthermore its located far from urban and industrial areas. The groundwater as the main water source for irrigation of this village.

The potential risk status of farmland had been studied previously around the world. In the Xunyang area, the mean value of RI was 259, indicating a moderate ecological risk, and Hg and Cd were the main heavy metal elements, which posed a high and considerable potential ecological risk, respectively, in the agricultural area [5]. On the Qinghai-Tibet Plateau, the average concentrations of Hg and Cd were 0.28 and 0.68 mg/kg, respectively, which exceeded their background value 40 and 108 times, respectively, and the RI value in the soil ranged from 234.6 to 375.9. Hg and Cd posed a high potential ecological risk for the region [22]. The soils in the Saudi Arabian dense agricultural area exhibited degrees of heavy metal potential ecological risk that ranged from large to small, and followed the ranking of Cd, Pb, Cu, Cr, Ni, Zn and Cd, the latter of which posed the highest potential ecological risk among the heavy metal elements [41]. In the north-eastern area of Hanoi, Vietnam, the agricultural soil presented a moderate environmental risk, and As and Cd had relatively high contribution rates [42]. In comparison, the potential ecological risk level [5], [22] and contribution order of the heavy metals [43–44] were anastomosis with former researchers' findings.

## 3.4 Source apportionment for the different elements by the PMF model

**3.4.1 Source apportionment.** To further identify and quantify the source of heavy metals in the suburban vegetable bases and to determine the contribution of each heavy metal, the PMF model (version 5.0) was used with eight parameters of soil samples. In our study, to ensure model fitness and determine the best solution, all heavy metals were defined as "strong" except for Pb, Hg and As, and the number of factors was set to 4 and 5. Then, the PMF model was run 20 times. Whether the number of factors was 4 or 5, the Q value was the smallest and the vast majority of the residuals was between -3 and 3, but the coefficient of the observed and predicted values ($R^2$) and the number of heavy metals was different. Thus, we considered the pollutant degree accounting for each heavy metal, selected five factors, and then running the PMF model. The relationship between the heavy metal concentrations and predicted concentrations is shown in Fig 4. Cr, Hg and As had high $R^2$ values of 0.96, 0.99 and 0.99, respectively, and Cu, Zn and Cd had $R^2$ values greater than 0.6. The remaining heavy metals, namely, Ni and Pb, had lower $R^2$ values of 0.57 and 0.54, respectively. While the $R^2$ values of Ni and Cd were slightly lower, the predicted values of Ni and Cd represented the observed values well at most sites, which did not affect finding the best solution. Therefore, the PMF model used a reasonable number of factors to better demonstrate the information contained in the original data.

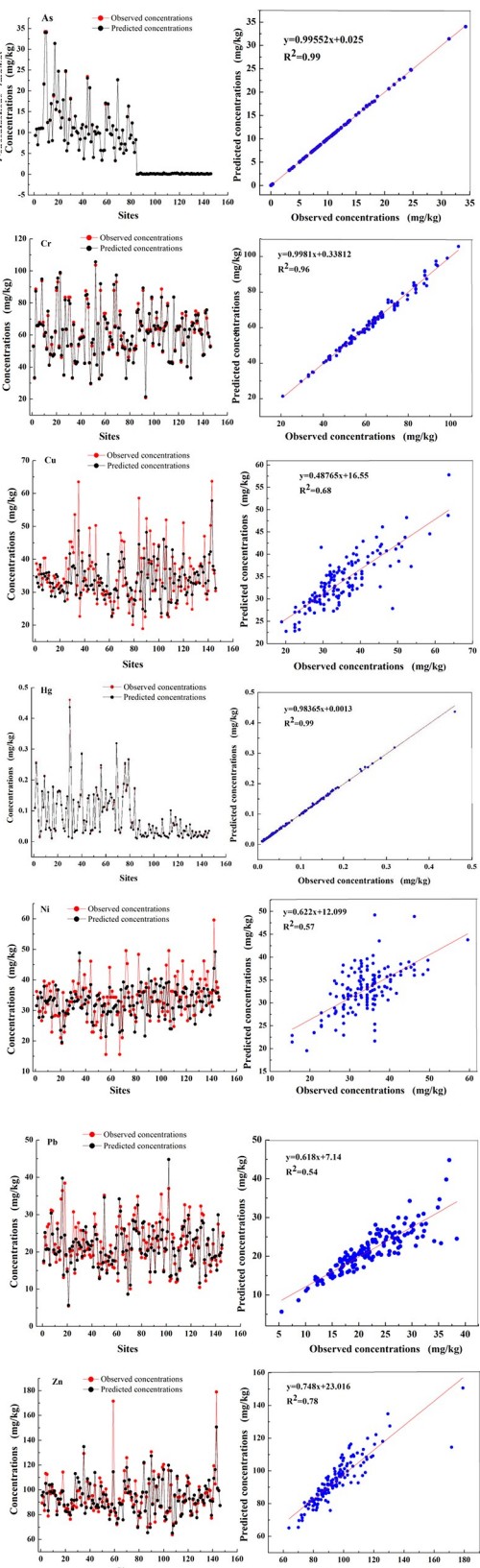

**Fig 4. Relationship between the observed and predicted heavy metal concentration by the PMF method.**

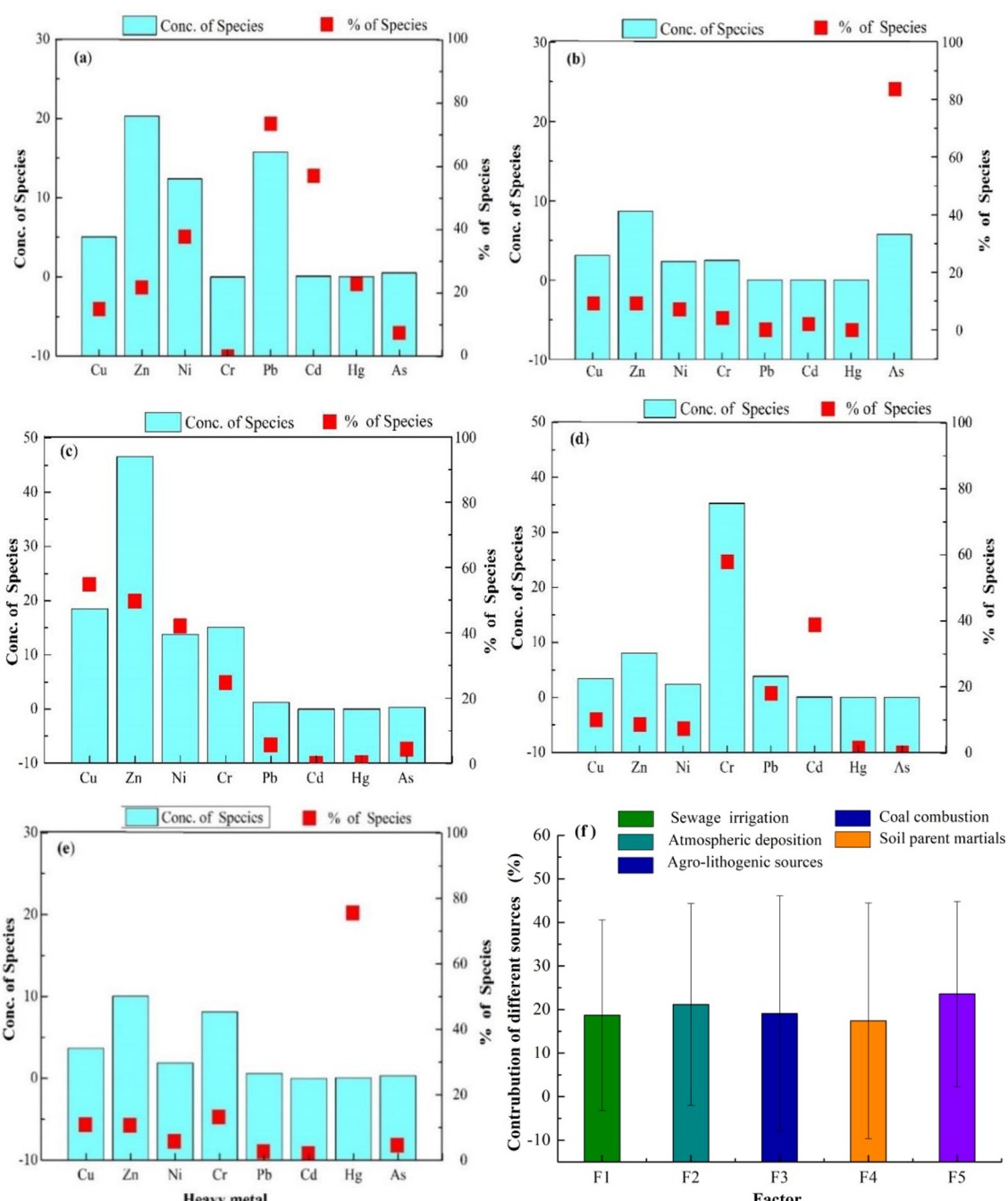

**Fig 5. Source profiles and source contributions of soil heavy metal from PMF.** (a. factor 1; b. factor 2; c. factor 3; d. factor 4; e. factor 5; f. the mean contribution rate (%) of 5 factors).

Factor 1 was mainly dominated by Pb and Cd (48.50 and 55.90%, respectively; Fig 5A). Moderate and high coefficients of variation could reflect the influence of humankind [45]. Both Pb and Cd had moderate coefficients of variation, 28.73 for Pb and 39.07 for Cd, which indicates that they were influenced by human activities (Table 2). Pb is an indicator element of traffic emission due to the utilization of engines and catalysts as well as the burning of fuels [46]. In addition, Huang et al. [47] found that most Pb in the environment came from traffic

emission using the Pb isotope ratio. Hu et al. [48] studied the adsorption and desorption characteristics in the soil in arid areas and found that oasis soil in arid areas had a great adsorption capacity for Pb. Moreover, there were two national highways (312 and 216) and two county highways (Tuwuda and Wukui), which crossed farmland, and the Pb discharged from motor vehicles accumulated in the topsoil causing Pb pollution of the soil. On the one hand, Cd mainly came from three industrial waste sources, namely, waste gas, wastewater and waste residue [49]. Li et al. [50] studied the spatial distribution and source identification of heavy metals in insoluble snow in Urumqi (China) and found that the contents of Cd in the northeast (Anningqu), where industrial areas were more common compared to other areas, exhibited a reduced tendency from the northeast to the southeast. In addition, some research also found that the concentration of Cd in dust in Urumqi (China) exceeded the local background value [51]. Furthermore, compared to the local background value, the mean concentration value of Cd exceeded that value by 1.45 times. Therefore, the first factor was classified as atmospheric deposition.

Factor 2 was characterized by Cr at a level of 69.80% (Fig 5B). The range of the Cr concentration was from 20.84 to 103.63 mg/kg, and the mean value was greater than the reference value, which indicated that the Cr in the soil in the vegetable bases, to a certain extent, had undergone human activities. In addition, some research found that the concentration of Cr in soil exceeded the background value by 1.38 times in China [52].The study area is located northeast of Urumqi, where nearby areas have many industrial factories, such as iron and steel thermo-electric factories, and is defined as an industrial area. Cr in the environment is immobile, less soluble and stable; thus, Crenters the soil environment via waste disposal emanating from a series of industrial activities, such as timber treatment, industrial metal processing and sewage sludge [53]. Increases in the Cr concentration are well known from sewage farm soils that are associated with long-term irrigation with wastewater [54]. Moreover, the study area has a long-term history of irrigation with sewage but currently stopped sewage irrigation. In a previous study, Cr in the irrigation water exceeded the background value by approximately 9 times [28]. Therefore, the second factor could be considered a sewage irrigation source.

Factor 3, As, had a high factor loading value of 82.90% (Fig 5C), which was significantly higher than that of the other remaining elements and was a marker element for factor three. While the coefficient of variation of As was 110.16%, which was the largest value among the eight heavy metals in soil, the average concentration of As (6.89 mg/kg) was lower than the reference value (11.2 mg/kg) by 1.63 times. Moreover, the percentage of uncontaminated sample sites was approximately 90% (Fig 2B), which means that the As content in the soil in the study area was relatively low. In summary, factor three was classified as the soil parent material.

Factor 4, Hg, received a higher weighting than the remaining heavy metals (83.60%, Fig 5D). The average concentration of Hg was 0.079 mg/kg (Table 2), which was greater than the reference value (0.017 mg/kg). This indicated that Hg pollution in vegetable soil had been increasing from anthropogenic sources. Tian et al. [55] found that the total emission of Hg from municipal solid waste burning in China significantly increased from 5.35 t in 2003 to 36.7 t in 2010. In addition, Lei et al. [56] showed that Hg mainly came from coal-fired power plants, non-ferrous metal smelting and cement production, and other sources, such as iron and steel production and residential coal combustion. For Xinjiang (China), the proportion of residential coal combustion was greater than the other sources. The study results of Zhang et al. [57] showed that the atmospheric deposition of Hg could change the Hg isotopic composition in the farmland topsoil close to industrial areas, and the Hg content in the topsoil was higher than that in the subsoil. The investigation reported that Hg content in coal in China is relatively high (about 0.17 mg/kg). In the process of burning coal, Hg can easily escape into the atmosphere with flue gas and fly ash, and then enters the soil through atmospheric deposition

[58]. Furthermore, Urumqi has a permanent population of 3.55 million people with a population density of 173.5 people/km$^2$ [25]. The duration of the winter in Urumqi is relatively long, and early October begins the heating season, which lasts until the middle march. According to field sampling, there were many greenhouses that were mainly used to produce vegetables in the winter using coal. In summary, considering the above considerations, the fourth factor could be attributed to a complex source of industrial and residential coal combustion. Certain studies published similar results [21].

Factor 5 was defined by Cu, Zn, Pb and Ni (45.30, 41.60, 36.30 and 50.70%, respectively; Fig 5E). The mean concentrations of these heavy metals (Cu, Zn, Pb and Ni) exceeded the respective reference values, indicating that these metals in the soil had accumulated to some degree. Almost all farmers in the study area were planting vegetables and applied a large amount of fertilizer (mineral fertilizer, organic fertilizer and manure) and pesticide to maintain and improve the soil fertility and ensure the production. Joimel et al. [59] studied heavy metals in French urban vegetable gardens and found that the concentration of Cu and Zn was higher than the local background value due to the utilization of fertilizers and pesticides. In addition, Cu and Zn are common ingredients in certain pesticides, and the use of pesticides is likely to generate a sum of 5000 t of Cu and 1200 t of Zn entering the soil per year in China [60]. Moreover, Cu, as an inherent component of additives in livestock diets, is transferred to animal manures, and thus Cu is normally a marker of the livestock manure applications [61–62]. The accumulation of Cu and Zn in soil was likely caused by manure utilization due to these heavy metals being present in the feedstuff as an additive against antimicrobial effects and for growth promotion [63]. Thus, the two heavy metals entered the soil via agricultural practices. On the other hand, Ni is mainly associated with trace elements such as Fe and Mn, which are probably derived from soil parent materials, and the Ni in the farmland soil is generated from subsequent pedogenesis [64]. While the concentrations of these heavy metals exceeded the reference values, more than 79% of the samples of the three metals were not contaminated (Fig 2B), the differences between the observed concentrations and reference values were not large, and the coefficients of the three heavy metals were not high (Table 2). Therefore, according to the low coefficient of variation and the high percentage of unpolluted sample sites, the anthropogenic inputs of Zn and Cu in fertilizers and pesticides may have been lower than the contents already present in the soil. In summary, the fifth factor could be attributed to the combination of an agricultural and lithogenic source. Certain other studies have reported similar results.

The mean contribution rates of the different pollution sources were calculated using the intensity of the samples and the source contribution of each heavy metal estimated by the PMF method and are presented in Fig 5F. It was found that the combined agricultural and lithogenic source had the greatest contribution (23.59%), followed by sewage irrigation (21.17%), soil parent material (19.11%), and atmospheric deposition (18.70%) as well as industrial and residential coal combustion (17.43%). Clearly, the soil contamination with heavy metals was mainly derived from an agro-lithogenic source and sewage irrigation, and the contribution rate of those sources accounted for 44.76%. This result, to a certain extent, illustrated whether the contributions of agricultural sources (fertilization and sewage irrigation), atmospheric depositions, or various industrial (coal combustion) activities to the heavy metal accumulation in soil were greater than those by natural sources (Fig 6). Therefore, to ensure the soil quality and quantity as well as green agricultural development, we could not ignore the various sources, specifically the agricultural sources. In addition, the agricultural and industrial activities would need to be properly adjusted and strictly limited in these vegetable bases.

**3.4.2 Spatial intensities of the sources.** The spatial distribution pattern of the different source intensities of the heavy metals in soil in the vegetable bases varied (Fig 6). The ranges of intensity of each factor were varied, for example, approximately 0.08–2.29 for factor 1,

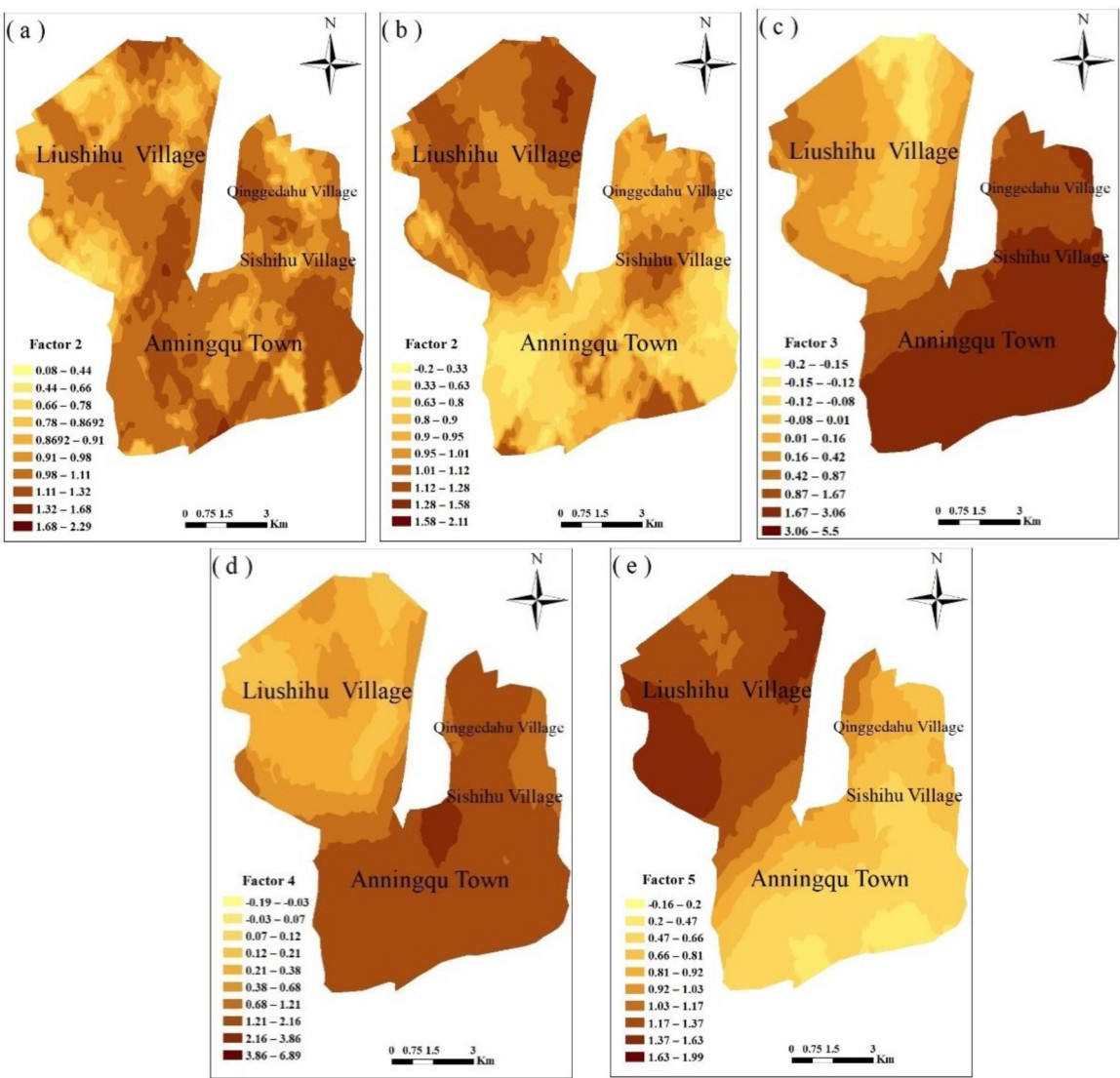

**Fig 6. Spatial distribution of the different source intensities of heavy metals in the soil.**

approximately -0.2–2.11 for factor 2, approximately -0.2–5.15 for factor 3, approximately -0.19–6.89 for factor 4 and approximately 0.16–1.99 for factor 5 (Fig 6). The mean intensity of each factor was the same (1.00), while the magnitudes of the intensity of each factor differed from one another.

There was no notable regular trend in the spatial distribution of the intensity of the atmospheric deposition. Higher and lower intensities of this factor were alternately distributed (Fig 6A). Higher intensities of sewage irrigation were concentrated in the northern part (Liushihu village) and north-eastern part (Sishihu village) of the vegetable bases, while the intensities in the southern part were relatively low (Fig 6B). It is a widespread and serious issue around the world that soils are being contaminated with heavy metals via sewage irrigation, for instance, in France [65], Germany [66], India [67] and China [68]. The vegetable bases have a long-term history of irrigation with effluent from sewage treatment plants, and Cr in the well water used for irrigation exceeded the standard limit [28]. The intensity of the soil parent material was

largely different in the study area. High intensities were mainly located in Anninqu town, Sishihu village and Qinggedahu village, while low intensities of this factor were distributed in Liushihu village (Fig 6C). The mean values of the As concentration, SD and CV for all soil samples were 6.89 mg/kg, 7.59 and 110.16%, respectively (Table 2). The mean values of the As content, SD and CV for the three areas (Anningqu town, Sishihu village and Qinggedahu village) in the vegetable bases were 11.91 mg/kg, 6.37 and 53.51%, respectively, and 0.09 mg/kg, 0.8 and 86.37%, respectively, for Liushihu village. When comparing the three mean As content values, the average As contents in Anningqu town, Sishihu village and Qinggedahu village were higher than the other two average concentrations and slightly exceeded the respective background values. There were no notable low intensities of residential and industrial coal combustion (Fig 6D). Higher intensities of this factor were mainly distributed in Anningqu town and Sishihu village. The intensity distribution pattern was consistent with the distribution pattern of the RI (Fig 3B). The fourth factor was mainly dominated by Hg in the soil (Fig 5D), and the Hg contribution of 259 to the potential ecological risk resulted in this factor being ranked as first and as the main heavy metal element causing ecological risk (Fig 3A). The intensities of the application of fertilizers and pesticides and of lithogenic sources gradually increased from the southern part to the northern part, from the eastern part to the western part as well. The intensities of the fifth factor in Liushihu village were greater than those in the other three villages (Fig 6E). The intensity distribution pattern, however, was completely different from the distribution pattern of the RI (Fig 3B). Fig 3B suggests that the lower values of the RI were mainly located in Liushihu village, but Fig 6E suggests the opposite distribution pattern. This phenomenon was most likely associated with the toxicities of the heavy metals.

## 4. Conclusions

In this study, various approaches including the geo-accumulation index and ecological risk index were used to investigate the lateral distribution of the heavy metal risk. Positive matrix factorization (PMF) analysis was conducted to better identify the possible sources of the heavy metals in the vegetable bases. The results of heavy metal concentrations showed that the average concentrations of the eight heavy metals in the soil exceeded the local respective background values, except for As. The geo-accumulation index ($I_{geo}$) was lower than zero for Cu, As, Cd, Zn, Ni, Cr and Pb, while the $I_{geo}$ of Hg was greater than one, which indicates that Hg was the main element that caused contamination in the vegetable bases. The mean value of the potential ecological risk index (RI) was 259.89 and resulted in a moderate ecological risk, in which the potential ecological coefficients ($E^i_j$) of Hg and Cd were 187.36 and 43.6, respectively. The $E^i_j$ values of these two heavy metals accounted for 88.87% of the total RI. In addition, the PMF analysis revealed that Pb and Cd were dominated by atmospheric deposition, with contributions of 48.50 and 55.90%, respectively; 48.50% of the Pb concentration was controlled by atmospheric deposition, and Pb from agro-lithogenic sources accounted for 36.30%. The Cd concentration of 55.90% was dominated by atmospheric deposition. Sewage irrigation sources accounted for 69.80% of the Cr content, and the remaining concentration was controlled by other pollution sources. Soil parent material contributed 82.90% of the As content. Industrial and residential coal combustion accounted for 83.60% of the Hg concentration, and the remaining concentration was attributed to other sources. Agricultural and lithogenic sources accounted for 45.30% of the Cu concentration, 41.60% of the Zn concentration and 50.70% of the Ni concentration. The results of the study should be used to control and reduce the heavy metal element inputs and ensure the safety and quality of vegetable production through regular monitoring, source control and integrated environmental management.

## Supporting information

**S1 Table. Descriptive statistics of soil heavy metals in different suburban farmlands (Unit: mg/kg).**
(DOCX)

## Author Contributions

**Conceptualization:** Balati Maihemuti.

**Data curation:** Mireadili Kuerban, Tuerxun Tuerhong.

**Formal analysis:** Mireadili Kuerban.

**Funding acquisition:** Balati Maihemuti.

**Investigation:** Yizaitiguli Waili.

**Methodology:** Mireadili Kuerban, Balati Maihemuti, Tuerxun Tuerhong.

**Project administration:** Balati Maihemuti.

**Supervision:** Balati Maihemuti.

**Validation:** Mireadili Kuerban.

**Writing – original draft:** Mireadili Kuerban.

**Writing – review & editing:** Balati Maihemuti.

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
