## [Decision Letter · Decision Letter 0]

27 Aug 2019

PONE-D-19-20453

Ecological Risk Assessment and Source Identification of Heavy Metal Pollution in Vegetable Bases of Urumqi, China, using the Positive Matrix Factorization (PMF) Method

PLOS ONE

Dear Dr. Balati Maihemutia,

Thank you for submitting your manuscript to PLOS ONE. After careful consideration, we feel that it has merit but does not fully meet PLOS ONE’s publication criteria as it currently stands. Therefore, we invite you to submit a revised version of the manuscript that addresses the points raised during the review process.

ACADEMIC EDITOR:The manuscript needs considerable revision before publication. The authors have cited Table 3 in line no 471, 514, but the manuscript contains only Table 1. The authors should carefully revise the manuscript. 

We would appreciate receiving your revised manuscript by October 4. To enhance the reproducibility of your results, we recommend that if applicable you deposit your laboratory protocols in protocols.io, where a protocol can be assigned its own identifier (DOI) such that it can be cited independently in the future. For instructions see: http://journals.plos.org/plosone/s/submission-guidelines#loc-laboratory-protocols

We look forward to receiving your revised manuscript.

Kind regards,

Sartaj Ahmad Bhat, Ph.D

Academic Editor

PLOS ONE

**Journal Requirements:**

2. In your Methods section, please provide additional location information of the collection sites, including geographic coordinates for the data set if available.

3. We note that  Figure(s) 1 & 3 in your submission contain [map/satellite] images which may be copyrighted. All PLOS content is published under the Creative Commons Attribution License (CC BY 4.0), which means that the manuscript, images, and Supporting Information files will be freely available online, and any third party is permitted to access, download, copy, distribute, and use these materials in any way, even commercially, with proper attribution. For these reasons, we cannot publish previously copyrighted maps or satellite images created using proprietary data, such as Google software (Google Maps, Street View, and Earth). For more information, see our copyright guidelines: http://journals.plos.org/plosone/s/licenses-and-copyright.

a) You may seek permission from the original copyright holder of Figure(s) [#] to publish the content specifically under the CC BY 4.0 license.  

**Comments to the Author**

1. Is the manuscript technically sound, and do the data support the conclusions?

Reviewer #1: Yes

Reviewer #2: Yes

2. Has the statistical analysis been performed appropriately and rigorously? 

Reviewer #1: Yes

Reviewer #2: Yes

3. Have the authors made all data underlying the findings in their manuscript fully available?

Reviewer #1: Yes

Reviewer #2: No

4. Is the manuscript presented in an intelligible fashion and written in standard English?

Reviewer #1: Yes

Reviewer #2: Yes

5. Review Comments to the Author

Reviewer #1: In reference to this manuscript, I consider that the theme is appropriate and within the scope of your journal, but there are too many errors and shortcomings in the work.

1. Please avoid using first person point-of-view terms (e.g., we, our, my, etc.)

2. Please rewrite Key words to highly summarize the main job of your article.

3. In lines 104-106, please check the terminology of source apportionment out twice.

4. In the Introduction section, you only mentioned the importance of receptor model (PMF). However, there is no implication about ecological risk assessment, which is very important in this work. Please involve it to your Introduction.

5. Please avoid irregular writing format like lines 145, 178 and 195, etc.

6. Why was the even lower value larger than the low value in lines 347-348?

7. In lines 425, authors said that the coefficient of variation of As was the largest value among the eight heavy metals, please discuss it in depth.

8. The conclusions must be written in accordance with the objectives and with the results obtained. Please improve this part.

Reviewer #2: The manuscript represents good quality of work. However, there are certain shortcomings which are to be incorporated to get the manuscript published in such reputed journal.

• First line of introduction is a repeat of same as in abstract. So it can be deleted in Introduction.

• The results of pH, Soil texture and SOM mentioned in Materials and Methods (Section 2.3) should be presented in Results and Discussion section and instead the methodology part should be made clearer. For example, How was SOM measured?

• The statement : ‘The results from several former researchers studies were listed in the Table S1’ . Table S1 could not be seen. If it is supplementary, it should be opened via link.

• In Line 390, 432, 471, 514, there is mention of Table.3, but the table does not exist in manuscript.

• In figure captions “5th and 95th and 5th and 95th percentiles of all data”, correct it as ‘5th and 95th percentiles of all data’,

• Although results are well written but discussion needs improvement. If other studies are mentioned in table S1 (which is missing here), these studies are also to be correlated to the present work.

Accepted only after minor revisions

6. PLOS authors have the option to publish the peer review history of their article (what does this mean?). If published, this will include your full peer review and any attached files.

Reviewer #1: No

Reviewer #2: No

---

## [Author Response · Author response to Decision Letter 0]

23 Oct 2019

Response to Reviewers

• Reviewer #1:

1. Please avoid using first person point-of-view terms (e.g., we, our, my, etc.)

Response: Thank you for your comments and suggestions. We had improved the expression of this paper, please see the revised paper for details.

2. Please rewrite Key words to highly summarize the main job of your article.

Response: Thank you for your suggestions. We had revised Key words of this paper, please see the revised paper for details (which are highlighted in blue).

2. In lines 104-106, please check the terminology of source apportionment out twice.

Response: Thank you for your advice. We checked the lines 104-106 of this paper and we couldn't find any problem as you said in this sentence (To control and prevent heavy metal pollution, the source identification and apportionment are very important, and the selection of a proper and effective model is essential for accurate results)

3. In the Introduction section, you only mentioned the importance of receptor model (PMF). However, there is no implication about ecological risk assessment, which is very important in this work. Please involve it to your Introduction.

Response: Thank you for your suggestions. We had revised introductions of this paper, please see the revised paper for details (which are highlighted in blue).

5. Please avoid irregular writing format like lines 145, 178 and 195, etc.

Response: Thank you for pointing out the mistake. We had revised and please see the revised paper for details (which are highlighted in blue).

6. Why was the even lower value larger than the low value in lines 347-348?

Response: Thank you very much for your comments. The RI values of the Liushihu village was lower than other town or village, because this village far from the industrial area and have a less population, Furthmore, groundwater (in this area groundwater less polluted than surface water) was the main source for irrigation of this village.

7. In lines 425, authors said that the coefficient of variation of As was the largest value among the eight heavy metals, please discuss it in depth.

Response: Thank you for your suggestions.

8. The conclusions must be written in accordance with the objectives and with the results obtained. Please improve this part.

Response: We are extremely thankful your valuable comments that have greatly improved the clarity of this article.

Reviewer #2: 

• First line of introduction is a repeat of same as in abstract. So it can be deleted in Introduction.

Response: Thank you for pointing out the mistake. It has now been corrected.

• The results of pH, Soil texture and SOM mentioned in Materials and Methods (Section 2.3) should be presented in Results and Discussion section and instead the methodology part should be made clearer. For example, How was SOM measured?

Response: Thanks a lot for this constructive comments and suggestions. The methods of the manuscript are refined as the reviewer commented. please see the revised paper for details (which are highlighted in red).

• The statement: ‘The results from several former researchers studies were listed in the Table S1’. Table S1 could not be seen. If it is supplementary, it should be opened via link.

Response: Thanks a lot for this valuable comment and pointing out the shortcoming. As reviewer suggested, the supplementary table take on opened via link.

• In Line 390, 432, 471, 514, there is mention of Table.3, but the table does not exist in manuscript.

Response: Thank you for pointing this mistake out. It has now been corrected. Please check the Line 390, 432, 471, 514 (which are highlighted in red).

• In figure captions “5th and 95th and 5th and 95th percentiles of all data”, correct it as ‘5th and 95th percentiles of all data’,

Response: Thank you for pointing out this mistake. It has been revised in figure caption list.

• Although results are well written but discussion needs improvement. If other studies are mentioned in table S1 (which is missing here), these studies are also to be correlated to the present work.

Response: we are very appreciated your valuable comments that have greatly improved the clarity of this article.

The editor’s suggestion and reviewer’s comments were really helpful and gave us a better perspective of our work. We hope that the revised manuscript would meet the requirement for publication. 

Thanks again for your kindly help.

Sincerely yours, 

Miradil Kurban, Balati Maihemuti, Ezzatgul Wali and Tuerxun Tuerhong

*Corresponding author: Balati Maihemuti

Address: College of Resource and Environment Sciences Xinjiang University, 14 Shengli Road, Tianshan Region, Urumqi 830046, People’s Republic of China 

E-mail addresses: bmaihemuti@xju.edu.cn

---

## [Decision Letter · Decision Letter 1]

2 Dec 2019

PONE-D-19-20453R1

Ecological Risk Assessment and Source Identification of Heavy Metal Pollution in Vegetable Bases of Urumqi, China, using the Positive Matrix Factorization (PMF) Method

PLOS ONE

Dear Balati Maihemuti,

Thank you for submitting your manuscript to PLOS ONE. After careful consideration, we feel that it has merit but does not fully meet PLOS ONE’s publication criteria as it currently stands. Therefore, we invite you to submit a revised version of the manuscript that addresses the points raised during the review process.

ACADEMIC EDITOR: I agree with the several very serious concerns of the Reviewer 1 who is in full agreement with the critics to this manuscript. Kindly revise the manuscript thoroughly. 

We would appreciate receiving your revised manuscript by Jan 16 2020 11:59PM. To enhance the reproducibility of your results, we recommend that if applicable you deposit your laboratory protocols in protocols.io, where a protocol can be assigned its own identifier (DOI) such that it can be cited independently in the future. For instructions see: http://journals.plos.org/plosone/s/submission-guidelines#loc-laboratory-protocols

We look forward to receiving your revised manuscript.

Kind regards,

Sartaj Ahmad Bhat, Ph.D

Academic Editor

PLOS ONE

Reviewers' comments:

Reviewer's Responses to Questions

**Comments to the Author**

1. If the authors have adequately addressed your comments raised in a previous round of review and you feel that this manuscript is now acceptable for publication, you may indicate that here to bypass the “Comments to the Author” section, enter your conflict of interest statement in the “Confidential to Editor” section, and submit your "Accept" recommendation.

Reviewer #1: All comments have been addressed

Reviewer #3: (No Response)

2. Is the manuscript technically sound, and do the data support the conclusions?

Reviewer #1: Partly

Reviewer #3: Yes

3. Has the statistical analysis been performed appropriately and rigorously? 

Reviewer #1: N/A

Reviewer #3: Yes

4. Have the authors made all data underlying the findings in their manuscript fully available?

Reviewer #1: No

Reviewer #3: Yes

5. Is the manuscript presented in an intelligible fashion and written in standard English?

Reviewer #1: No

Reviewer #3: Yes

6. Review Comments to the Author

Reviewer #1: Most of the comments that I sent previously were not answered/considered. Some problems still laid on this work, the first-person point-of-view terms are still used in this article, the key words are still cannot stand for this work, and the discussions are still in poor situation. Therefore, this article cannot be considered by this journal currently.

Reviewer #3: The research work presented in manuscript is important considering in increase in Pollution and thus heavy metals in biosphere. Though, some minor revisions are still required for the manuscript to be ready for publication.

• Use of Pronouns should be avoided in manuscript.

• Language needs improvement in the entire manuscript. Some sentences are incomplete, while others are not grammatically correct.

• Are you sure this is the first report on ecological risk associated with heavy metals in the vegetable bases, as mentioned in introduction? Or this is the first such report for the particular area?

• Probably you used SPSS 19.0 and not SPPS 19.0 as mentioned in line 242, please check.

• The sentence “The low (approx. 0-150) and even lower (approx. 150-300) RI values…….” Is still not clear. It can be rewritten as “in Liushihu village, where the density of human 351 activities and the amount of transportation were relatively low, groundwater was the 352 main source for irrigation and it was located far from urban and industrial areas, RI values exhibited light and moderate risk. Please modify.

7. PLOS authors have the option to publish the peer review history of their article (what does this mean?). If published, this will include your full peer review and any attached files.

Reviewer #1: No

Reviewer #3: No

---

## [Author Response · Author response to Decision Letter 1]

24 Feb 2020

We at the first place are very much thankful to you for giving us an opportunity to revise our manuscript according to the comments of the reviewers. Also we wish to express our heartfelt thanks to you for your systematic and scientific arrangement of peer review process. We would also like to thank the current three reviewers for their time spent in reviewing our manuscript and for their constructive comments. We have addressed all the points raised by the reviewers and also consequently revised the manuscript (the highlighted parts) as per the reviewers’ comments. The revision also has been proofread by native English speaker with the language editing service of American journal experts. Definitely based on their comments, we feel that the scientific quality of manuscript has been considerably improved to be selected for publication in PLOS ONE. We have great pleasure in submitting our revision with the response to the reviewers’ comments for your kind perusal and consideration.

---

## [Editor Report · Decision Letter 2]

25 Feb 2020

Ecological Risk Assessment and Source Identification of Heavy Metal Pollution in Vegetable Bases of Urumqi, China, using the Positive Matrix Factorization (PMF) Method

PONE-D-19-20453R2

Dear Dr. Maihemuti,

We are pleased to inform you that your manuscript has been judged scientifically suitable for publication and will be formally accepted for publication once it complies with all outstanding technical requirements.

With kind regards,

Sartaj Ahmad Bhat, Ph.D

Academic Editor

PLOS ONE
---

## [Editor Report · Acceptance letter]

23 Mar 2020

PONE-D-19-20453R2 

Ecological Risk Assessment and Source Identification of Heavy Metal Pollution in Vegetable Bases of Urumqi, China, using the Positive Matrix Factorization (PMF) Method 

Dear Dr. Maihemuti:

I am pleased to inform you that your manuscript has been deemed suitable for publication in PLOS ONE. Congratulations! Your manuscript is now with our production department. 

With kind regards,

on behalf of

Dr. Sartaj Ahmad Bhat 

Academic Editor

PLOS ONE